# Commonsense Reasoning with Implicit Knowledge in Natural Language

**Pratyay Banerjee**[*†]                                   pbanerj6@asu.edu
**Swaroop Mishra**[*†]                                     srmishr1@asu.edu
**Kuntal Kumar Pal**[*†]                                   kkpal@asu.edu
**Arindam Mitra**[*‡]                              arindam.mitra@microsoft.com
**Chitta Baral**[†]                                         chitta@asu.edu
[†]*School of Computing and AI, Arizona State University*
[‡]*Microsoft*

## Abstract

Commonsense Reasoning is a research challenge studied from the early days of AI. In recent years, several natural language QA task have been proposed where commonsense reasoning is important. Two common approaches to this are (i) Use of well-structured commonsense present in knowledge graphs, and (ii) Use of progressively larger transformer language models. While acquiring and representing commonsense in a formal representation is challenging in approach (i), approach (ii) gets more and more resource-intensive. In this work, we take a middle ground where we use smaller language models together with a relatively smaller but targeted natural language text corpora. The advantages of such an approach is that it is less resource intensive and yet at the same time it can use unstructured text corpora. We define different unstructured commonsense knowledge sources, explore three strategies for knowledge incorporation, and propose four methods competitive to state-of-the-art methods to reason with implicit commonsense.

## 1. Introduction

For an AI agent to reason about the everyday routine human activities, the agent needs to possess commonsense. Consequently, commonsense acquisition and reasoning are considered critical research challenges from the early days of AI [McCarthy, 1959]. The need for commonsense reasoning is reemphasized recently [Sap et al., 2019a, Marcus and Davis, 2019], particularly in NL understanding and QA. Several commonsense reasoning tasks have been proposed that study the different aspects of commonsense reasoning, such as abductive commonsense [Bhagavatula et al., 2019], physical commonsense [Bisk et al., 2019], and social commonsense [Sap et al., 2019b]. QA systems approach solving tasks using large-pretrained transformers, such as BERT [Devlin et al., 2019], or use complex knowledge fusion methods to perform QA [Lin et al., 2019, Lv et al., 2020].

In this paper, focusing on low resource use, we evaluate the use of smaller transformer language models and a small number of knowledge-rich natural language sentences, where relevant knowledge may be implicitly expressed. To understand what we mean by implicit knowledge, consider an example from [Winograd, 1972]: Given the context "*The city councilmen refused the demonstrators a permit because they feared violence.*", and the question "*Who is fearing violence?*", the correct answer is "*The city councilmen*". An

---

∗. Equal contribution

unstructured retrieved (through a web search engine) knowledge [Prakash et al., 2019] for this context-question pair that can help answer this question correctly is: "*He also refused to give his full name because he feared for his safety.*". We can use this knowledge to reason that the person who is refusing, is the one who is fearing. From this example, we can observe that the necessary commonsense knowledge to reason may be present in text in many cases but in an implicit way. Moreover, this knowledge is unstructured, and hence current state-of-the-art knowledge fusion methods are unable to utilize this knowledge without a method to represent it in a knowledge graph triple, as present in *ConceptNet*.

Using natural language sentences (as a source of knowledge) at first glance appears similar to the application of evidence retrieval for open-domain question answering [Yang et al., 2018, Clark et al., 2018, Kwiatkowski et al., 2019], where systems retrieve supporting evidence to be able to answer an open-ended question. However, there is a big difference as, unlike in evidence retrieval, the needed commonsense knowledge may not be *explicitly* available in unstructured knowledge corpora. Our approach is to reason-with-example, in contrast to reading comprehension with retrieved supporting paragraphs containing answers or explicit knowledge that lead to answers. Moreover, a high lexical overlap with a retrieved knowledge and context-question-answer does not mean it can be used to answer correctly. For example, another retrieved knowledge for the above question is: "*Demonstrators fear the retaliatory police violence.*". An additional layer of complexity to commonsense reasoning with natural language is added because of such high lexical overlap but distracting sentences.

We limit our study to two pre-trained transformers, namely BERT and RoBERTa. BERT and RoBERTa have been trained using 13GB and 160GB data, respectively. RoBERTa has the same architecture and parameter count but is trained with extensive hyper-parameter tuning and has a larger vocabulary (25K v/s 50K). These allow us to study the implicit commonsense reasoning ability with varying pre-training and vocabulary size. Larger pre-trained transformers have been effectively shown to improve performance on downstream tasks, but training such models is resource-intensive. Hence we ask the following auxiliary question: To what extent can we improve a smaller transformer encoder's performance? Smaller in the sense of pre-training data, vocabulary size, and parameter tuning space.

For addressing the above questions, we propose the following experimental framework. We categorize different unstructured knowledge sources and define a knowledge source preparation and retrieval component. We then propose three strategies of unstructured knowledge infusion. In the *Revision strategy*, we fine-tune the transformer on an unstructured knowledge source. In *Openbook strategy*, we choose a certain number of knowledge statements from the unstructured knowledge source that are textually similar to each of the dataset samples. Then we fine-tune the pre-trained transformer for the question-answering task. In the final strategy, we combine both the strategies mentioned above. We propose three strong baseline methods that utilize knowledge, *concat*, *max*, *simple-sum*, and an explainable reasoning model *weighted-sum* to combine and reason with multiple commonsense knowledge sentences. We evaluate our proposed framework on three public commonsense question answering datasets: AbductiveNLI (aNLI) [Bhagavatula et al., 2019], PIQA [Bisk et al., 2019] and Social Interaction QA (SIQA) [Sap et al., 2019b].

Our key findings are as follows: (a) Transformers can reason with implicit commonsense knowledge to some extent. We observe that transformers fail to answer questions through detailed error analysis even when sufficient knowledge is present with minimal distractors

| Abductive NLI | Social IQA | Physical IQA |
|---|---|---|
| **Obs1**: Jim was working on a project. 
 ✓ Jim found he was missing an item. 
 ✗ Jim needed a certain animal for it. 
 **Obs2**: Luckily, he found it on a nearby shelf 
 **Knowledge**: Peyton eventually found it before Peyton needed to determine that something is missing. Kendall never found it, as a result Kendall wants to lodge a missing complaint. | **Context**: Remy was an expert fisherman and was on the water with Kai. Remy baited Kai's hook. 
 **Question**: What will Remy want to do next? 
 ✓ cast the line 
 ✗ put the boat in the water 
 ✗ invite Kai out on the boat 
 **Knowledge**: Alex baits Pat's hook as a result others want to cast their line. | **Goal**: When doing sit-ups: 
 ✓ place your tongue in the roof of your mouth. It will stop you from straining your neck. 
 ✗ place your elbow in the roof of of your mouth. It will stop you from straining your neck. 

 **Knowledge**: How to Do Superbrain Yoga. Place your tongue on the roof of your mouth. |

Figure 1: Example of all three datasets along with retrieved knowledge.

30-50% of the time. This observation shows the scope of future improvements. (b) Revision and Openbook Strategy improve commonsense reasoning performance, but the Revision strategy's impact depends on how well-formed the unstructured knowledge corpus is. (c) Our knowledge retrieval and knowledge infusion methods improve accuracy over pre-trained transformers by 2-9%. They are significantly effective over the smaller transformer encoders and approach larger pre-trained transformers, surpassing T5-11B [Raffel et al., 2019] by 4.14% in aNLI and reducing the gap to 1.75% in SIQA using RoBERTa. These methods should act as future baselines.

In summary, our contributions are: (a) a thorough analysis of transformers' ability to perform commonsense reasoning with implicit knowledge on three different commonsense QA tasks using two transformer models. (b) four models representing four ways knowledge can be infused in transformer encoders. These methods apply to multiple commonsense reasoning tasks and improve performance over pre-trained transformers by 2-9% in accuracy. (c) a detailed study to bridge the gap between smaller and larger pre-trained transformers. (d) an extensive investigation to study the impact of different knowledge sources and pre-training on such knowledge sources on commonsense QA tasks.

## 2. MCQ Datasets

To study the extent of transformers' commonsense reasoning ability, we choose the following three datasets to evaluate our models, each with a different kind of commonsense knowledge. Figure 1 shows examples from each of the datasets with our retrieved commonsense knowledge sentences.

**Abductive NLI (aNLI):** This dataset [Bhagavatula et al., 2019] is intended to judge the potential of an AI system to do abductive reasoning to form possible explanations for a given set of observations. The task is to find which of the hypothesis options $H_1$, and $H_2$ explains $O_2$ where $O_1$ should precede and $O_2$ should succeed the hypothesis, given a pair of observations $O_1$ and $O_2$. This task needs a commonsense understanding of which order sequence of events occurs. There are 169,654 train and 1,532 validation samples. The test set is blind. It has a generation task, but we restrict ourselves to the multiple-choice task.

**PIQA (Physical Interaction QA):** This dataset is created to evaluate an AI system's physics reasoning capability. The dataset requires reasoning about physical objects and how we use them in our daily lives. The task is to predict the most appropriate choice to the goal $G$, given a goal $G$ and a pair of choices $C_1$ and $C_2$. There are 16,113 train and 1,838 validation samples. The test set is blind.

**SIQA:** This dataset is a collection of instances about social interaction reasoning and the social implications of their statements. The task is to choose the correct answer option $AO_i$ out of three choices when given a context $C$ of a social situation and a question $Q$ about the situation. There are several question types in this dataset derived from *Atomic* inference dimensions [Sap et al., 2019a]. A few of the Atomic inference dimensions are actor *intention*, actor *motivation*, *effect* on the actor, *effect* on others, etc. In total, there are 33,410 train and 1,954 validation samples. The test set is blind.

## 3. Commonsense Knowledge Sources

### 3.1 Knowledge Categorization for Evaluation

**Directly Derived:** Here the commonsense QA task is directly derived from the knowledge source, and hence using the same knowledge may make the task trivial. We test this scenario on the aNLI task with the following knowledge sources, *ROCStories Corpus* [Mostafazadeh et al., 2016] and *Story Cloze Test*, that were used in creating aNLI. Our motivation is to see how well the model can answer questions when given the "same" or similar implicit/explicit commonsense knowledge.

**Partially Derived:** Here the commonsense QA task is not directly derived from an external knowledge source, and considerable human knowledge was used to generate the question-answers. In this case, we use SIQA as the task, which uses the *Atomic* [Sap et al., 2019a] knowledge base as the source for social events, but has undergone sufficient human intervention to make the task non-trivial. During dataset creation, the human annotators were asked to turn *Atomic* events into sentences and were asked to create question-answers.

**Relevant:** Here, the commonsense task is entirely created with human annotators' help without using a specific knowledge source. However, we guess knowledge sources that seem relevant through our QA pairs analysis. We evaluate this using PIQA as the commonsense task and *WikiHow* dataset [Koupaee and Wang, 2018] as the "relevant" external knowledge source.

### 3.2 Knowledge Source Preparation

**aNLI:** To test our first category of external knowledge, we use the entire *Story Cloze Test* and *ROCStories Corpus*. We also prepare another source that contains knowledge sentences retrieved for the train set of aNLI from the first source. This knowledge source is created to ensure the task is not trivialized with knowledge leakage. We also create a knowledge source from multiple datasets such as *MCTest* [Richardson et al., 2013], *COPA* [Roemmele et al., 2011] and *Atomic*, but not *Story Cloze Test* and *ROCStories Corpus*. These sources contain commonsense knowledge, which might be useful for the aNLI task.

**SIQA:** We synthetically generate a knowledge source from the events and inference dimensions provided by the *Atomic* dataset [Sap et al., 2019a]. The *Atomic* dataset contains events and eight types of if-then inferences *. The total number of events is 732,723. Some events are masked, which we fill by using a BERT and masked language modeling [Devlin et al., 2019]. We extend the knowledge source, and replace *PersonX* and *PersonY*, as present in

---

*. More details in Supplemental Materials.

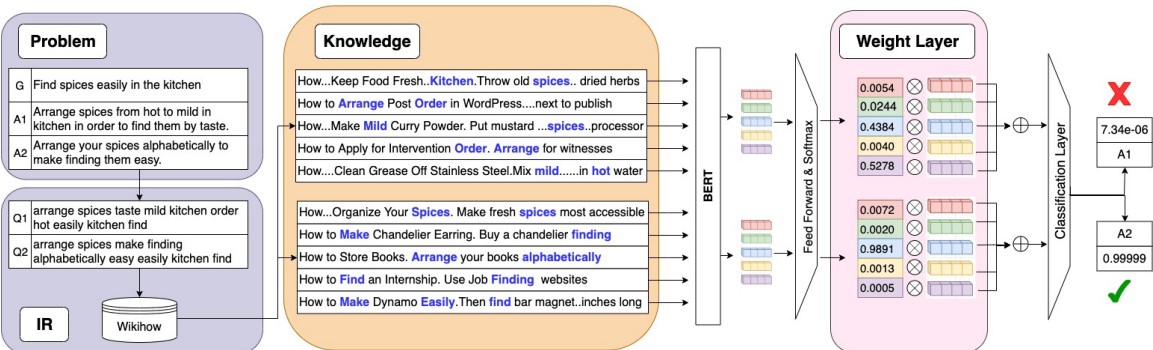

Figure 2: An end-to-end view of our approach. From query generation, knowledge retrieval, the different types of knowledge retrieved along with keywords highlighted in blue, the corresponding learned weights in the Weighted-Sum model and finally to predicted logits.

the original *Atomic* dataset, using gender-neutral names. These steps may approximate the steps taken by humans to generate QA pairs.

**PIQA:** We use the *Wikihow* dataset for PIQA. It contains paragraphs (214,544) with detailed steps or actions to complete a task. We extract the title of each paragraph and split the paragraphs into sentences. The title is concatenated to each of the sentences. This preprocessing ensures that the task's goal is present in each of the sentences.

A Combined Commonsense Corpus is created which combines the partially related and relevant corpuses, for example, combining *Wikihow*, *Atomic*, *MCTest*.

### 3.3 Knowledge Retrieval

**Query Generation:** We concatenate the question, answer option, and the context if present, and remove standard English stopwords for query generation. We use common nouns, verbs, adjectives, and adverbs from the QA pairs. Explicit bias towards specific names (John, Jane) is avoided.

**Information Retrieval System:** We use Elasticsearch to index all knowledge base sentences. We retrieve the top 50 sentences for each QA pair with the default BM-25 ranking model [Robertson and Walker, 1994]. The retrieved sentences may contain the key search words in any order.

**Re-Ranking:** We re-rank the retrieved knowledge sentences to remove redundant sentences containing the same information. We use sentence similarity and knowledge redundancy to perform the iterative re-ranking. We use Spacy, to compute cosine similarity between sentence Glove vector [Pennington et al., 2014] representations; for knowledge redundancy, we find similarity with the already selected sentences and discard a new sentence if it is $> 0.9$ similar to higher-ranked sentences. After re-ranking, we select the **top ten** sentences.

We keep our Information Retrieval system generic as the tasks require varying kinds of commonsense knowledge; for example, If-then rules in SIQA, Scripts or Stories in aNLI, and understanding of Processes and Tools in PIQA.

| Dataset | Strategy | BERT | | | | RoBERTa | | | |
|---|---|---|---|---|---|---|---|---|---|
| | | Concat | Max | Sim–Sum | Wtd–Sum | Concat | Max | Sim–Sum | Wtd–Sum |
| **aNLI** | OPENBOOK | 73.9± 0.8 | 73.7± 0.1 | 73.5± 0.7 | 73.3± 1.0 | 83.9± 0.5 | 80.8± 0.9 | 81.7± 0.6 | 84.4± 0.4 |
| | REVISION | 72.7± 0.3 | N/A | N/A | N/A | 82.4 | N/A | N/A | N/A |
| | REVISION & OPENBOOK | 74.4± 0.2 | 74.3± 0.1 | 74.0± 0.9 | 75.1±0.4 | 84.2± 0.7 | 81.4± 0.8 | 82.6± 0.6 | **86.7**± 0.6 |
| **PIQA** | OPENBOOK | 67.8± 0.4 | 72.4± 0.6 | 72.6± 1.2 | 72.5± 0.1 | 74.8± 0.5 | 75.2± 0.9 | 75.6± 0.7 | 77.1± 0.2 |
| | REVISION | 74.5± 0.3 | N/A | N/A | N/A | 75.2± 0.8 | N/A | N/A | N/A |
| | REVISION & OPENBOOK | 67.7± 0.1 | 73.8± 0.8 | 76.8± 0.5 | 76.8± 0.3 | 75.4± 0.7 | 76.2± 0.8 | 76.8± 0.4 | **80.2**± 0.6 |
| **SIQA** | OPENBOOK | 70.1± 0.8 | 67.8± 0.1 | 70.0± 0.7 | 70.2± 0.4 | 76.5± 0.7 | 77.2± 0.6 | 77.4± 0.2 | 78.3± 0.5 |
| | REVISION | 69.5± 0.9 | N/A | N/A | N/A | 76.8± 0.3 | N/A | N/A | N/A |
| | REVISION & OPENBOOK | 68.8± 0.4 | 66.6± 0.4 | 68.9± 0.1 | 69.3± 0.6 | 78.2± 0.3 | 77.4± 0.9 | 76.7± 0.5 | **79.5**± 0.9 |

Table 1: Validation set accuracy (%) of each of the four models (Concat, Max, Simple sum, Weighted sum). Revision only method has no retrieved passage, so only Q-A is concatenated.

## 4. Method

After extracting relevant knowledge from the respective KBs, we move onto the task of Question-Answering. We perform our experiments on BERT encoders, with 340M and 355M parameters respectively, BERT-Large (Low vocab-size 25K and pretraining data 13GB) BERT [Devlin et al., 2019] and RoBERTa (high-vocab size 50K and pretraining data 160 GB ) RoBERTa [Liu et al., 2019].

**QA-Model:** As a baseline, we use these pre-trained transformers for the question answering task with an extra feed-forward layer for classification as a fine-tuning step.

### 4.1 Modes of Knowledge Infusion

We experiment with four different models of using knowledge with the transformer architecture for the open-book strategy. The first three, *concat*, *max*, and *simple-sum* act as stronger baselines that use the same implicit knowledge as our proposed *weighted-sum* model. Each of these modules takes as input a problem instance which contains a question $Q$, $n$ answer choices $a_1, ..., a_n$ and a list called *premises* of length $n$, one for each answer. Each element in *premises* contains $m$ number of knowledge passages, which might be useful while answering the question $Q$. Let $K_{ij}$ denotes the $j$ th knowledge passage for the $i$ th answer option. Each model computes a score of $score(i)$ for each of the $n$ answer choices. The final answer is the answer choice that receives the maximum score. We now describe how the different models compute the scores differently.

**Concat:** In this model, all the $m$ knowledge passages for the $i$-th choice are joined together to make a single knowledge passage $K_i$. The sequence of tokens {[CLS] $K_i$ [S] $Qa_i$ [S]} is then passed to BERT to pool the [CLS] embedding ($z^{[CLS]}$) from the last layer. This way we get $n$ $z^{[CLS]}$ for $n$ answer choices, each of which is projected to a real number ($\texttt{score}(i)$) using a linear layer.

**Parallel-Max:** For each answer choice $a_i$, Parallel-Max uses each of the knowledge passage $K_{ij}$ to create the sequence {[CLS] $K_{ij}$ [S] $Qa_i$ [S]} which is then passed to the BERT model to obtain the $z^{[CLS]}$ from the last layer that is then projected to a real number using a linear layer. $\texttt{score}(i)$ is the max of the $m$ scores obtained using each of the $m$ knowledge passage.

**Simple Sum:** In *simple sum* and the next model assumes that the information is scattered over multiple knowledge passages and try to aggregate that scattered information. To do this, the *simple sum* model, for each answer choice $a_i$ and each of the knowledge passage

| Models/ Accuracy | aNLI | | PIQA | | SIQA | |
|---|---|---|---|---|---|---|
| | Val | Test | Val | Test | Val | Test |
| **BERT** | 67.36 | 66.75 | 68.08 | 69.23 | 64.88 | 64.50 |
| **GPT-2 XL** | N/A | N/A | 70.20 | 69.50 | 47.50 | 45.30 |
| **RoBERTa** | 85.05 | 83.91 | 76.28 | 76.80 | 77.85 | 76.74 |
| **RoBERTa 5 Ensemble** | N/A | 83.22 | N/A | 79.66 | N/A | 78.68 |
| $L2R^2$ [2020] | N/A | **86.81** | N/A | N/A | N/A | N/A |
| **KagNet** [2019] | N/A | N/A | N/A | N/A | 65.05 | 64.59 |
| **GBR** [2020] | N/A | N/A | N/A | N/A | 75.64 | 76.25 |
| **UnifiedQA T5 11B** [2020] | N/A | 80.04 | N/A | **89.50** | N/A | **79.75** |
| **Ours: BERT + WS** | 74.60 | 74.96 | 76.82 | 72.28 | 70.21 | 67.22 |
| **Ours: RoBERTa + WS** | 85.90 | 84.18 | 80.20 | 78.24 | 79.53 | 78.00 |

Table 2: Performance of the Weighted-Sum model with *Revision* & *Openbook* strategy, compared to current best methods. Underlined are methods that we beat statistically significantly. Partially derived and related sources are used. Unavailable→N/A. Best→Bold.

$K_{ij}$ creates the sequence {[CLS] $K_{ij}$ [S] $Qa_i$ [S]} which it then passes to the BERT model to obtain the $z^{[CLS]}$ from the last layer. All of these $m$ vectors are then summed to find the summary vector, projected to a scalar using a linear layer to obtain the `score`$(i)$.

**Weighted Sum:** The *weighted sum* model computes a weighted sum of the $m$ $z^{[CLS]}$ as some of the knowledge passage might be more useful than others. It computes the $z^{[CLS]}$ in a similar way to that of the *simple sum* model. It computes a scalar weight $w_{ij}$ for each of the $m$ $z^{[CLS]}$ using a linear projection layer which we will call as the *weight layer*. The weights are then normalized through a softmax layer and used to compute the weighted sum of the $z^{[CLS]}$. It then uses (1) a linear layer or (2) reuses the weight layer (*tied version*) to compute the final score `score`$(i)$ for the option $a_i$. We experiment with both options.

Formally, given $m$ $z^{[CLS]}$, we learn two projections $w_1$ and $w_2$, such that:

$$score(i) = w_2( \sum_{j=1}^{n} w_1(z^{[CLS]}) * z^{[CLS]}) \tag{1}$$

This weighted-sum of vectors is similar to the attention weights learned to create contextual word vectors [Vaswani et al., 2017] but we extend it to multiple sentences. We minimize the cross-entropy loss between the score and the ground-truth answer. We observe a single layer network achieves the best accuracy compared to multi-layer feed-forward networks and highway networks for projection.

## 5. Experiments

Let $D$ be an MCQ dataset, and $T$ be a pre-trained transformer, $K_D$ be a knowledge source (a set of paragraphs or sentences) which is useful for $D$ and let $K$ be a general knowledge source where $T$ was pre-trained, and $K$ might or might not contain $K_D$. We consider three approaches to infuse knowledge.

**Revision:** In this strategy, $T$ is fine-tuned on $K_D$ using Masked LM (both BERT and RoBERTa) and the next sentence prediction task (BERT) and then fine-tuned on the dataset $D$ for the QA task.

**Openbook:** Here a subset of $K_D$ is assigned to each of the training samples in the dataset $D$ as a knowledge passage context, and the model $T$ is fine-tuned on the modified dataset $D$.

| Model | Knowledge Source | aNLI | PIQA | SIQA |
|---|---|---|---|---|
| **BERT** | Directly/Partially Derived | 75.1± 0.4 | N/A | 70.2± 0.4 |
| | TrainOnly Directly/Partially | 74.6± 0.8 | N/A | 69.8± 0.7 |
| | Related Knowledge | 73.2± 0.5 | 76.8± 0.3 | 68.6± 0.5 |
| **RoBERTa** | Directly/Partially Derived | 86.7± 0.6 | N/A | 79.5± 0.9 |
| | TrainOnly Directly/Partially | 85.9± 0.8 | N/A | 78.9± 1.2 |
| | Related Knowledge | 85.0± 1.1 | 80.2± 0.6 | 77.4± 0.8 |

Table 3: Effect of different knowledge sources types on the Weighted-Sum knowledge infused model. Related Knowledge source is the combination of all relevant knowledge sources, referred to as the Combined Commonsense Corpus. Metric is Accuracy.

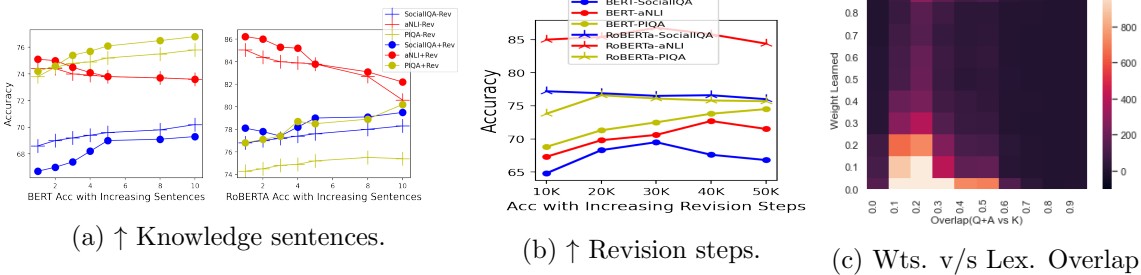

(a) ↑ Knowledge sentences.

(b) ↑ Revision steps.

(c) Wts. v/s Lex. Overlap

Figure 3: For **a**, **b**, and **c** the knowledge infusion model is Weighted-Sum with knowledge retrieved from a relevant knowledge source. In Fig. **a**, we observe the effect of increasing number of implicit knowledge sentences. In Fig. **b** we observe the effect of increasing number of *Revision* pre-training steps. Fig. **c** shows the weights learned vs. normalized lexical overlap between knowledge and concatenated QA pair for all samples of PIQA dev set.

**Revision with an Openbook:** In this strategy, $T$ is fine-tuned on $K_D$ using Masked LM (both BERT and RoBERTa) and the next sentence prediction task (BERT) and also a subset of $K_D$ is assigned to each of the training samples on $D$. The model is then fine-tuned for the modified dataset $D$.

We train the models on 4 Nvidia V100 16GB GPUs with learning rates in the range [1e-6,5e-5] and batch sizes of [16,32,48,64]. We report the mean accuracy for three random seed runs. We perform five hyper-parameter trials and param-selection on the validation set.

## 6. Results and Discussion

Tables 1, 2 and 3 summarize our results on three datasets. BERT and RoBERTa baseline validation and hidden test scores are present in Table 2. Adding knowledge in natural language form improves QA accuracy statistically significantly across all datasets over the baseline BERT with $p \leq 0.05$ based on Wilson score intervals [Wilson, 1927]. This includes retrieving knowledge from related knowledge sources, seen in Tables 2 and 3. The *concat* mode of knowledge infusion improves over the baseline BERT by 1-6%, and the Weighted-Sum model further improves it by 2-4%. In Table 2 we can observe the Weighted-Sum model is 4.1% better than T5 in aNLI and reduces the gap to 1.75% in SIQA with 30 times less number of parameters (11B v/s 355M). It also surpasses complex graph-based approaches like GBR and KagNet [Lin et al., 2019, Lv et al., 2020]. Other prior work use

| Strategy | Training Src. | aNLI | SIQA | PIQA |
|---|---|---|---|---|
| **OpenBook** | **aNLI** | N/A | 63.2 65.5 | 51.2 57.8 |
| | **SIQA** | 72.4 84.1 | N/A | 48.5 54.3 |
| | **PIQA** | 62.5 74.2 | 49.6 54.2 | N/A |
| **Revision** | **aNLI** | N/A | 65.3 66.2 | 56.2 65.8 |
| | **SIQA** | 70.9 83.8 | N/A | 52.4 57.8 |
| | **PIQA** | 66.1 78.0 | 57.4 67.6 | N/A |
| **OpenBook + Revision** | **aNLI** | N/A | 65.8 68.2 | 55.4 62.8 |
| | **SIQA** | 73.1 85.2 | N/A | 53.2 59.4 |
| | **PIQA** | 63.8 75.6 | 52.8 63.1 | N/A |

Table 4: Effect of cross-dataset knowledge source accuracy on Weighted-Sum (when a relevant source for a different task is used). BERT Left, RoBERTa Right.

directly derived knowledge sources and model for specific tasks as in L2R [Zhu et al., 2020]. Moreover, UnifiedQA T5 11B [Khashabi et al., 2020] is trained on many datasets, whereas we train only on the provided train dataset, making our approach more sample efficient. This observation validates our hypothesis of using implicit knowledge expressed in natural language to bridge the gap to super-large transformers. Our generic framework improves on all three datasets with models trained only using the provided training dataset.

**Effect of different strategies:** Both the *Openbook* and the *Revision* strategies perform well. Together the performance improves even further. The performance of the *Revision* strategy is low for SIQA. The drop in performance may be attributed to the sentences' synthetic nature and the unavailability of next sentence prediction task data, as the knowledge in the KB for SIQA is single sentences and not paragraphs. PIQA and aNLI results are better due to natural and contiguous sentences. For PIQA, the BERT model improves with knowledge, whereas the RoBERTa model underperforms, indicating RoBERTa gets distracted by the retrieved knowledge, and the pre-training knowledge is more useful. BERT with implicit knowledge approaches RoBERTa without knowledge, with the gap reduced by 4% on average. Similarly, RoBERTa approaches T5 with *Revision* & *Openbook* strategy.

**Effect of different knowledge sources:** Table 3 shows the impact of different knowledge sources on the downstream question answering task. Even a knowledge source with somewhat related knowledge is impactful for the question answering task, as seen in the case of Related Knowledge and TrainOnly Partially Derived for aNLI and SIQA. In Directly and Partially derived knowledge categories, such as RoCStories for aNLI and *Atomic* for SIQA, the model accuracy with knowledge is significantly more than the baseline but does not reach near-human accuracy. However, the model can still not answer all questions because the model fails to reason well even with sufficient knowledge, and the annotators have modified the information present in the source knowledge significantly. As a result, the knowledge does not overlap with the gold answer, cause if it did, the model will use lexical overlap as a short-cut and perform better. In Table 4, we can observe aNLI and SIQA require similar commonsense knowledge, as training with the relevant knowledge source of aNLI has a non-detrimental effect for SIQA and vice-versa. We also observe PIQA performance decreases if we use a knowledge source of aNLI and PIQA, indicating it introduces a significant amount of distraction such that even the implicit knowledge in pre-trained transformers is ignored. [†]

---

† More details and the error analysis are in Supplemental Materials.

| Knowledge | aNLI | SIQA | PIQA |
|---|---|---|---|
| **Explicitly Present** | 14% | 11% | 10% |
| **Implicitly Present** | 55% | 59% | 51% |
| **Fully Irrelevant** | 31% | 30% | 39% |

| Types of Error | aNLI | SIQA | PIQA |
|---|---|---|---|
| **Annotation** | 41% | 38% | 10% |
| **Model Prediction** | 48% | 27% | 29% |
| **Distracting Knowledge** | 11% | 35% | 61% |

Table 5: Left: Percent of correct predictions where the implicit knowledge is categorized as above, for the RoBERTa Weighted-Sum model. Right: Different types of errors observed in the QA pairs where the RoBERTa Weighted-Sum model failed to answer correctly.

**Comparisons between modes of knowledge fusion:** The Weighted-Sum model is observed to be consistent across datasets. The other strong baseline models also improve over the no-knowledge models indicating even simple scoring methods over implicit commonsense knowledge sentences can lead to improvements. The Max, Simple-Sum, and Weighted-Sum models have an additional advantage of being partially explainable by observing the weights associated with the knowledge sentences. Weighted-Sum outperforms them as it has the flexibility to attend in varying degrees to multiple sentences, in contrast to other models. Figure 2 shows the weight versus overlap between knowledge and QA pair distribution for PIQA. There is an overall low overlap, but the model learns to give high weights regardless of the overlap. It indicates that the model captures the implicit knowledge and not just a simple word overlap. We observe 61% of such low lexical overlap sentences have sufficient implicit knowledge on manual analysis.

**Why the impact of external knowledge is less for RoBERTa?** RoBERTa has been pre-trained using a gigantic corpus of 160 GB text. We assume for these tasks that the model needs additional knowledge to answer, but we hypothesize that the pre-training corpus of RoBERTa might contain the knowledge we are trying to infuse, leading to the reduced impact. This observation calls for further analysis of pre-training corpora to categorize such commonsense knowledge. The significant improvement over BERT (3-14%) shows the ability for these methods to utilize implicit knowledge, which is especially useful for low-resource languages, target domains where we can pre-train using fewer data and use ad-hoc knowledge to solve a target task and have smaller vocab and params. But, there is an assumption that atleast sufficient data (~10GB) to train a BERT model is necessary. Future work will explore the size v/s knowledge impact for even smaller language models.

**Error Analysis:** We analyzed 200 correct predictions and error samples from each of our best models, respectively. In Table 6, we can observe for around two-third of the correct predictions, we have relevant knowledge present. The model also ignores partial noise by reducing its weight and the entire knowledge passage if needed. In those cases, we hypothesize that the knowledge acquired during the revision phase or the original language model pre-training phase helps answer correctly. We divide the errors into three categories, as seen in Table 6. *Annotation Errors* are when more than one answer option is correct, or an incorrect answer option is labeled correctly. The questions for which information is insufficient to select a specific answer option also fall into this category. *Distracting knowledge* is where the retrieved knowledge is noisy and does not have sufficient relevant knowledge. *Model prediction* error is where the relevant knowledge is present, though the knowledge is not wholly exact. However, a human could have reasoned with the provided knowledge.

## 7. Related Work

**Commonsense Reasoning:** Several attempts were made to inject external knowledge into neural networks to improve commonsense QA in recent years. A knowledge selection algorithm to rank knowledge paths from *ConceptNet* via PMI and frequency-based scoring was proposed by Bauer et al. [2018]. Wang and Jiang [2019] improve word representations by integrating common word vectors between document and question-answer options. A commonsense-based pre-training was proposed by Zhong et al. [2019] to learn direct and indirect *ConceptNet* relations. Lin et al. [2019] proposed a knowledge-augmented graph-based reasoner and pruning knowledge paths using a function adapted from a graph embedding algorithm. Lv et al. [2020] is the closest work that utilizes both a structured knowledge base and explicit unstructured plain text as a source to enhance contextual representations. Our Revision strategy is similar to task adaptive pre-training, but we focus on commonsense knowledge infusion, whereas Gururangan et al. [2020] focuses on textual domain adaptation for text classification.

**Transformers Reasoning Abilities:** Recently, a few attempts were made to understand the different reasoning abilities of transformer models. Clark et al. [2020] observe that transformers can reason with explicit conjunctive implication rules and observe a strong performance. Talmor et al. [2020] study to what extent transformers can reason over explicit symbolic facts while retaining implicit pre-training knowledge. Richardson and Sabharwal [2020] study if the transformer QA models know definitions and taxonomic reasonings and propose probing datasets. Gontier et al. [2020] study the ability to generate proofs given knowledge encoded in natural language. In contrast to the above studies, we study the ability to reason with additional implicit commonsense knowledge [‡].

## 8. Conclusion

In this work, we comprehensively study transformers' ability to reason with implicit knowledge expressed in natural language. We propose an experimental framework with knowledge infusion methods and observe a considerable improvement of 2-9% over strong baselines. We observe our methods, trained with fewer samples and parameters, perform competitively with huge pre-trained language models and surpass complex graph-based methods [Lin et al., 2019, Lv et al., 2020]. Moreover, the approaches we studied are general enough to apply to other knowledge-intensive tasks and languages. Our methods reduce the gap between smaller and large pre-trained transformers. We critically analyze the different components and identify that transformers are still unable to answer 30-50% of the time, even with sufficient knowledge, identifying the need for better methods to perform reasoning with implicit knowledge. We hope our findings will help design models that respond better to instructions [Mishra et al., 2021] containing knowledge expressed in natural language.

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

## 9. Supplemental Material

In the first section, we analyze BERT predictions for all three datasets. In the second section we analyze prediction errors unique to each datasets . The third section is regarding analysis of our dataset aNLI, where we have attempt to find out why the model is not able to answer all questions even with the help of directly derived knowledge source. In the third section, we compare RoBERTa and BERT by analyzing the weighted-sum model. Fourth section illustrates some examples in these datasets which need external facts beyond common sense knowledge.

### 9.1 Detailed Related Work

**Question-Answering Datasets:** In datasets such as SQuAD [Rajpurkar et al., 2016], TriviaQA [Joshi et al., 2017], WikiQA [Yang et al., 2015], CoQA [Reddy et al., 2019], HotpotQA [Yang et al., 2018] the answers are present in either the passage or the context, or require multi-hop reasoning and retrieval. Another challenging QA task is when the multiple-choice questions do not have sufficient knowledge to answer correctly given a passage, context or options, as in ARC [Clark et al., 2018], RACE [Lai et al., 2017], and OpenBook QA [Mihaylov et al., 2018]. Numerical commonsense reasoning datasets [Mishra et al., 2020c, Lin et al., 2020] demand models to know various numerical facts and thus poses challenges. Recently language models trained on a huge corpus have been able to perform quite well [Devlin et al., 2019, Liu et al., 2019] on them. Our focus in this paper is on datasets that require external facts and need *commonsense knowledge* to predict the correct answer, as in aNLI, PIQA, and SIQA.

**Commonsense Reasoning:** Several attempts were made to inject external knowledge into neural networks to improve commonsense QA in recent years. A knowledge selection algorithm to rank knowledge paths from *ConceptNet* via PMI and frequency-based scoring was proposed by Bauer et al. [2018]. Wang and Jiang [2019] improve word representations by integrating common word vectors between document and question-answer options. A commonsense-based pre-training was proposed by Zhong et al. [2019] to learn direct and indirect *ConceptNet* relations. Lin et al. [2019] proposed a knowledge-augmented graph-based reasoner and pruning knowledge paths using a function adapted from a graph embedding algorithm. Lv et al. [2020] is the closest work that utilizes both a structured knowledge base and explicit unstructured plain text as a source to enhance contextual representations. Systems that use structured knowledge sources aim to utilize relational knowledge of the form $(a, R, b)$, where $a$ and $b$ are words, to modify pre-trained word vectors in both passages and questions to obtain better inter-word alignments. In our case, knowledge is more involved, with $a$ and $b$ being event descriptions containing variables. We move away from using knowledge graphs and focus on implicit knowledge sentences.

**Transformers Reasoning Abilities:** Recently, a few attempts were made to understand the different reasoning abilities of transformer models. Clark et al. [2020] observe that transformers can reason with explicit conjunctive implication rules and observe a strong performance. Talmor et al. [2020] study to what extent transformers can reason over explicit symbolic facts while retaining implicit pre-training knowledge. Richardson and Sabharwal [2020] study if the transformer QA models know definitions and taxonomic reasonings and propose probing datasets. Gontier et al. [2020] study the ability to generate proofs given

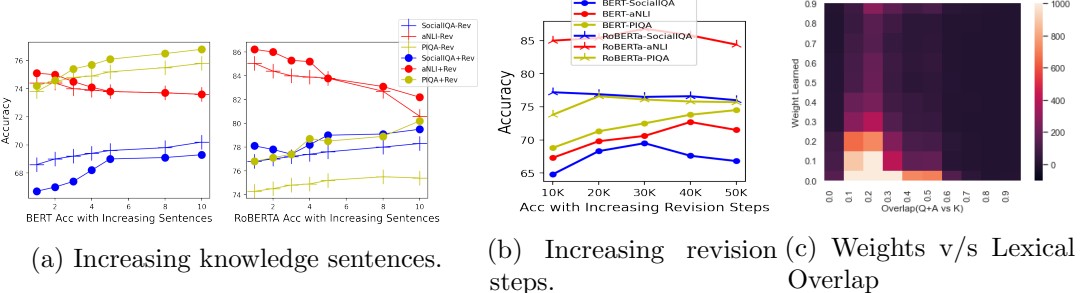

(a) Increasing knowledge sentences.

(b) Increasing revision steps.

(c) Weights v/s Lexical Overlap

Figure 4: For **a**, **b**, and **c** the knowledge infusion model is Weighted-Sum with knowledge retrieved from a relevant knowledge source. In Fig. **a**, we observe the effect of increasing number of implicit knowledge sentences. In Fig. **b** we observe the effect of increasing number of *Revision* pre-training steps. Fig. **c** shows the weights learned vs. normalized lexical overlap between knowledge and concatenated QA pair for all samples of PIQA validation set.

knowledge encoded in natural language. In contrast to the above studies, we study the ability to reason with additional implicit commonsense knowledge.

**External Knowledge in QA:** Systems for evidence retrieval, such as Elasticsearch [Gormley and Tong, 2015], has been used in prior work of [Pirtoaca et al., 2019, Yadav et al., 2019, Banerjee et al., 2019, Banerjee and Baral, 2020a] . Other complex systems using supervised and unsupervised retrieval neural models over structured and unstructured knowledge sources are proposed for multihop reasoning and open-domain QA [Asai et al., 2019, Das et al., 2019, Lee et al., 2019, Lewis et al., 2020, Banerjee, 2019] . We use Elasticsearch for retrieval in our work, and we have an unsupervised re-ranking algorithm using Spacy [Honnibal and Montani, 2017]. Gururangan et al. [2020] has shown the need for task adaptive pre-training to improve target task performance. Our Revision strategy is similar to task adaptive pre-training, but we focus on commonsense knowledge infusion, whereas Gururangan et al. [2020] focuses on textual domain adaptation for text classification. Recent work use knowledge graphs and unstructured text to perform unsupervised question answering [Banerjee and Baral, 2020b, Ma et al., 2021].

### 9.2 Model Response to Variation in Number of Knowledge Sentences:

**Effect of number of sentences & revision steps:** In Figure 4, we observe a trend of increasing the number of sentences improving SIQA and PIQA accuracy but not for aNLI. In contrast, increasing revision steps improve aNLI and PIQA accuracy but not for SIQA for BERT and has a minor impact on RoBERTa. The negative trend in aNLI for increasing sentences is due to more distracting sentences. We are limited to 10 sentences as we are restricted with the maximum token size of 512.

### 9.3 Error Analysis

We analyzed 200 correct predictions and error samples from each of our best models, respectively. In Table 6, we can observe for around two-third of the correct predictions,

| Knowledge | aNLI | SIQA | PIQA |
|---|---|---|---|
| **Explicitly Present** | 14% | 11% | 10% |
| **Implicitly Present** | 55% | 59% | 51% |
| **Fully Irrelevant** | 31% | 30% | 39% |

Table 6: Percent of correct predictions where the implicit knowledge is categorized as above, for the RoBERTa Weighted-Sum model.

| Types of Error | aNLI | SIQA | PIQA |
|---|---|---|---|
| **Annotation** | 41% | 38% | 10% |
| **Model Prediction** | 48% | 27% | 29% |
| **Distracting Knowledge** | 11% | 35% | 61% |

Table 7: Different types of errors observed in the question-answer pairs where the RoBERTa Weighted-Sum model failed to answer correctly.

we have relevant knowledge present. The model also ignores partial noise by reducing its weight and the entire knowledge passage if necessary. In those cases, we hypothesize that the knowledge acquired during the revision phase or the original language model pre-training phase helps answer correctly.

We divide the errors into three categories, as seen in Table 7. *Annotation Errors* are when more than one answer option is correct, or an incorrect answer option is labeled correctly. The questions for which information is insufficient to select a specific answer option also fall into this category. *Distracting knowledge* is where the retrieved knowledge is noisy and does not have sufficient relevant knowledge. *Model prediction* error is where the relevant knowledge is present, though the knowledge is not wholly exact. However, a human could have reasoned with the provided knowledge.

Selection of 200 representative samples of each dataset to conduct an error analysis [Mishra et al., 2020b, Swayamdipta et al., 2020, Mishra and Sachdeva, 2020], and further incorporation of instance difficulty [Mishra and Arunkumar, 2021, Varshney et al., 2020, Mishra et al., 2020a] in the analysis, can provide more accurate insights on the intricacies of how models leverage knowledge provided in natural language.

We also have performed several dataset-specific error analysis which we explain below.

### 9.3.1 SOCIALIQA

We measure the performance across the eight different ATOMIC inference dimensions for the best knowledge infused model. The six of the inferential dimensions are Needs, Attributes, Reactions, Wants, Motivations, Effects. These are for PersonX. There are two more for Others, Reaction and Wants.

In figure 5 we can see that, both model with knowledge and model without knowledge performs nearly equally across all six dimensions without any considerable improvement across any particular dimension.

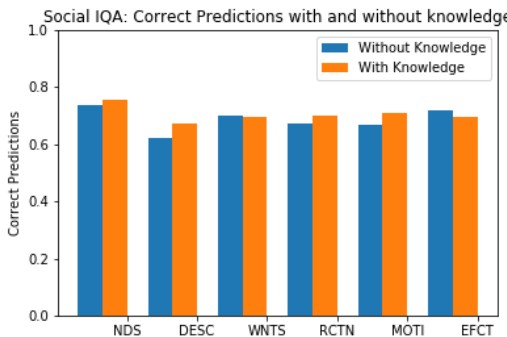

Figure 5: Performance of the model with (MAC model) and without knowledge (Baseline) across different types of ATOMIC inference dimensions.

In some cases the model fails to predict the correct answer despite of the appropriate knowledge being provided as shown below.

**Question:** *Kendall took their dog to the new dog park in the neighborhood. . What will Kendall want to do next?*
(A) *walk the dog* (B) **meet other dog owners Knowledge:** Jody takes Jody's dog to the dog park, as a result Jody wants to socialize with other dog owners.

341 questions were predicted wrongly after addition of external knowledge. We also identified out of the set of 100 analyzed correct predictions, 29% of the questions had partial information relevant to the question.

### 9.3.2 PIQA

Out of the 100 failures in PIQA, we find that for 8 samples the *goal* matches the knowledge statements but the answers present in the knowledge is different. This shows that our model fails for those 8 questions where there can be multiple correct answers. For example,

**Goal:** *How can I soothe my tongue if I burn it?*
(A) Put some salt on it. **(B) Put some sugar on it.**
**Knowledge Retrieved:** How to Soothe a Burnt Tongue.Chew a menthol chewing gum.

Also, there are 33 samples in the whole train and dev dataset for which, the words in one options are a subset of second option. In those cases, the knowledge retrieved is same for both the options and this confuses our models. For examples,

**Goal:** *What can I drink wine out of if I don't have a wine glass?*
(A) Just pour the wine into a regular mug or glass and drink. **(B) Just pour the wine into a regular mug or wine glass and drink.**
**Knowledge Retrieved:** How to Serve Foie Gras. Pour a glass of wine.

With incorporation of knowledge, 359 samples which were initially incorrectly predicted by our models without knowledge, is now correctly predicted for PIQA dataset. On the other hand this leads to 166 samples which were correct have now become incorrect.

## 9.3.3 ANLI

In this dataset, we also have some examples where negative knowledge is being fed to the model, and it still produces the correct output. There are 8 such examples among the 100 samples we analyzed. For example:

---

**Obs1:** *Pablo likes to eat worms.*
**Obs2:** *Pablo does not enjoy eating worms.*
(Hyp1) Pablo thought that worms were a delicious source of protein. **(Hyp2) Pablo then learned what worms really are.**

**Knowledge:** Pablo likes to eat worms. He read a book in school on how to do this. He fries them in olive oil. He likes to do this at least once a month. Pablo enjoys worms and views them as a delicacy.

---

Similarly, we have examples where knowledge favors incorrect hypothesis, however our system still produces correct output. We found 12 such examples among the 100 samples we analyzed. For example:

---

**Obs1:** *Dotty was being very grumpy.*
**Obs2:** *She felt much better afterwards.*
(Hyp1) Dotty ate something bad. **(Hyp2) Dotty call some close friends to chat.**
**Knowledge:** Allie felt not so good last night. She ate too much. So she had to sleep it off. Then she woke up. She felt so much better

---

We have 12 cases among 100 analyzed samples, where both hypothesis are very similar. So, our system is unable to produce correct output. For example:

---

**Obs1:** *Bob's parents grounded him.*
**Obs2:** *He came back home but his parents didn't even know he left.*
(Hyp1) Bob got caught sneaking out. **(Hyp2) Bob got away with sneaking out.**

---

We have 34 examples where incorrect hypothesis has more word similarity with the observation and knowledge, whereas correct hypothesis has been paraphrased or has less raw word similarity. The system predicts the wrong answer in such a situation. One such example is:

---

**Obs1:** *Mary's mom came home with more bananas than they could possibly eat.*
**Obs2:** *That was the best way ever to eat a banana!*
**(Hyp1) Mary and her mom decided to make chocolate covered frozen bananas to avoid waste.** (Hyp2) Mary made pineapple splits for everyone.
**Knowledge:** Mary s mom came home with more bananas than they could possibly eat. She wondered why she had bought them all. Then after dinner that night she got a surprise. Mom made banana splits for the whole family. That was the best way ever to eat a banana

---

Another area where the system fails, is where the problem seems to be open-ended, and many hypotheses can explain the pair of observations. It is tough to find exact knowledge

in such a scenario. For example,

---

**Obs1:** *Lisa went for her routine bike ride.*
**Obs2:** *Some days turn out to be great adventures.*
**(Hyp1) Lisa spotted a cat and followed it off trail** (Hyp2) Lisa saw a lot of great food.
**Knowledge:** Lisa went for her routine bike ride.Only this time she noticed an abandoned house.She stopped to look in the house.It was full of amazing old antiques.Some days turn out to be great adventures.

---

### 9.4 Why Language model is not able to answer all questions even with the help of directly derived knowledge source?

We experiment by providing directly derived knowledge source in aNLI, and find that the language model is still not able to answer all questions. We analyze and find following insights regarding aNLI and direcly derived knowledge source.

1. 41.90 % of the data has either Obs1 or Obs2 or both common with the knowledge.

2. 23.12 % of the data has both Obs1 and Obs2 common with the knowledge.

Among those, Figure 6 illustrates percentage overlap of hypothesis with knowledge. Overlap has been calculated by taking set of words in hypothsis, and finding if the same word is there in knowledge or not. You can see that very few examples have high overlap with knowledge. Actual number will be even lesser, as we have calculated word overlap, not phrase or sentence overlap.

For each of the bins, Figure 7 illustrates how much of those are correctly classified. It is interesting to see that, even though knowledge is very much relevant here (obs1, obs2 same and high hypothesis overlap), BERT has not been able to classify correctly. Bins 0-10 and 10-20 can be ignored here, since percentage of data in those bins are very less as seen from Figure 6.

### 9.5 BERT prediction analysis

To understand how knowledge is being utilized by BERT and whether the knowledge is actually useful or not, we randomly select 100 samples where our best performing model predicts correctly and 100 samples where it fails.

For those *100 correctly predicted samples*, we consider four categories : (1) Exact appropriate knowledge is present, (2) A related but relevant knowledge is present, (3) Knowledge is present only in the correct option, and (4) No knowledge is present. Figure 8 shows the counts for the above categories. All the cases do not occur in all the datasets. This analysis helps us in understanding for how many of the samples, the model is actually using knowledge.

For those *100 samples where our model fails*, (Figure 9), we analyze, (1) Is the knowledge insufficient, (2) Is the knowledge support the wrong answer, (3) Is the Knowledge is appropriate but still our model fails, and finally (4) Gold label is questionable.

We also analyze given appropriate knowledge, how the model performs. From Figure 8, it can be seen that BERT can answer quite a number of question without knowledge. Also

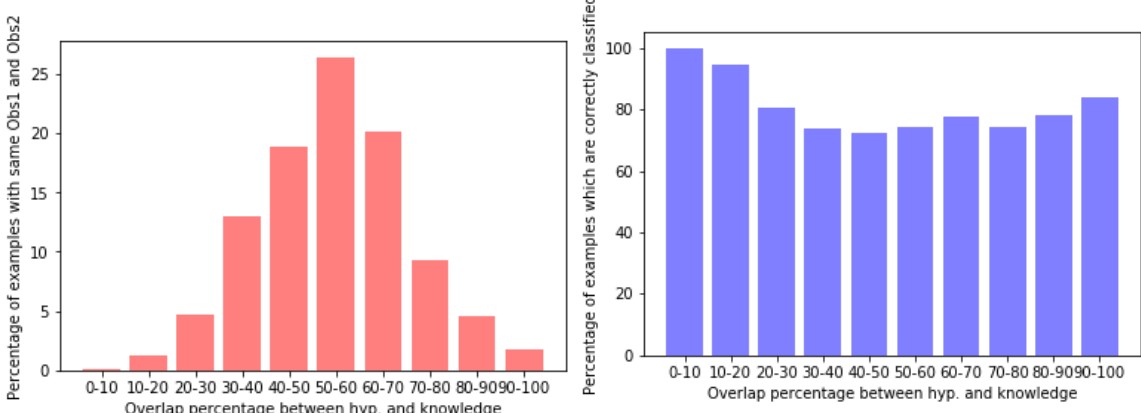

Figure 6: Measure of performance across different knowledge presence in correct predictions

Figure 7: Measure of performance across different knowledge presence in incorrect predictions.

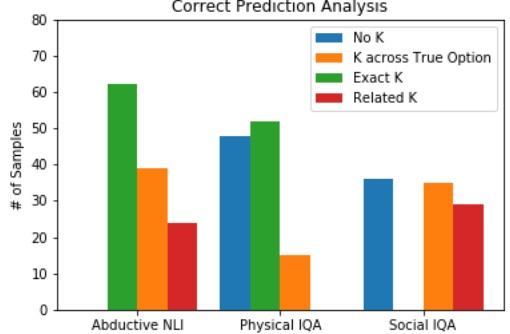

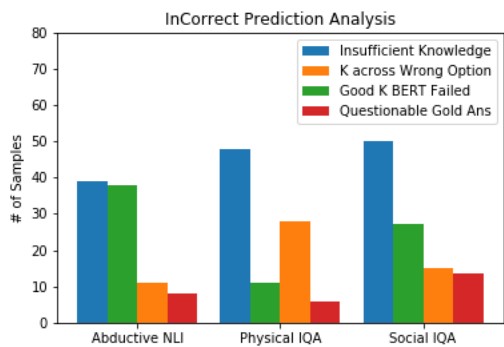

Figure 8: Measure of performance across different knowledge presence in correct predictions

Figure 9: Measure of performance across different knowledge presence in incorrect predictions.

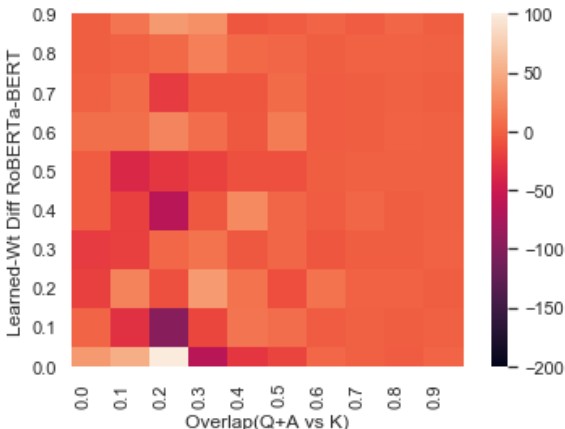

Figure 10: Difference between weights learned by the RoBERTa Weighted-Sum Model vs BERT-Weighted-Sum Model for the Normalized Overlap between knowledge and concatenated question-answer for all samples of PIQA validation set

from Figure 9, it is clear that despite of having appropriate knowledge, in some cases BERT fails to answer correctly.

### 9.6 RoBERTa vs BERT

In the Figure 10, both the learned weights and the percentage overlap between the question and option versus knowledge is binned with an interval of 0.1. The figure shows the difference between the counts of samples for each weight and overlap bin between BERT-weighted-sum model and RoBERTa-weighted-sum model. From the figure, it can be seen that, the samples with percentage overlap between 0.2 to 0.3 which have been assigned lower weight in the region 0.0 to 0.1 are more in BERT weighted sum model than in RoBERTa weighted sum model.

On the other hand, the samples with percentage overlap between 0.2 to 0.3 which have been assigned weight in the region 0.1 to 0.2 and in region 0.2-0.3 are more in RoBERTa weighted sum model than BERT weighted sum model. This shows the RoBERTa model is able to assign weights to proper knowledge sentences leading to improved question answering performance.

### 9.7 Examples which need external facts beyond commonsense knowledge

There are some questions which need external facts beyond common sense knowledge to answer. Following are two examples.

| |
|---|
| **Goal:** how do you call the fuzz? 
 **(A) dial 911.** 
 *(B) dial fuzz under contacts.* |
| **Goal:** To fight Ivan Drago in Rocky for sega master system.? 
 *(A) Drago isn't in this game because it was released before Rocky IV.* 
 **(B) You have to defeat Apollo Creed and Clubber Lang first.** |

