# OpenReview forum: "Commonsense Reasoning with Implicit Knowledge in Natural Language"
_AKBC.ws/2021/Conference — AKBC 2021_

### Official Review · Reviewer_urSv · 2021-07-20
**A solid work with detailed empirical analysis.**

**Rating:** 7
**Confidence:** 5

**Review:**

The paper presents a detailed study on the commonsense reasoning ability of pre-trained Transformer-based LMs such as BERT and RoBERTa. The authors build an experimental framework for evaluating different models with different knowledge sources under different infusion strategies, e.g., continually pre-training (revision) or retrieval-based learning (openbook), or both. The authors present a few methods that can improve the results and reduce the gap between smaller LMs and larger ones.

Questions and suggestions

- I suggest the authors consider adding the CommonsenseQA task [1] and an additional dataset and use the OMCS corpus (the text-version of ConceptNet) or GenericsKB [2] as the knowledge resource. Also, the OpenCSR [3] seems to share the similar goal of the openbook setup for answering commonsense questions.

- The authors claim in the conclusion that the proposed methods "surpass complex graph-based methods"? I don't find the supporting facts from the experiments.

- I would suggest the authors do not claim the methods are part of the novelty of the paper as they are very trivial. I would rephrase the Introduction and Section 4.1 so that the key contribution is the analysis and the evaluation protocol. The proposed methods here are just some default options as baselines in the framework.

- The experiment section does not show any qualitative analysis or case studies. It is unclear how well the retrieved sentences are and how they can improve the LMs for reasoning.

- The retrieval module can be replaced with pre-trained DPR [4], which is generally better than BM25.

[1] CommonsenseQA: A Question Answering Challenge Targeting Commonsense Knowledge.
Alon Talmor, Jonathan Herzig, Nicholas Lourie, Jonathan Berant. NAACL-19

[2] https://allenai.org/data/genericskb

[3] Differentiable Open-Ended Commonsense Reasoning.
Bill Yuchen Lin, Haitian Sun, Bhuwan Dhingra, Manzil Zaheer, Xiang Ren, William W. Cohen. NAACL-21

[4] Dense Passage Retrieval for Open-Domain Question Answering. EMNLP 2020

---

> ### Author Response · Authors · 2021-07-30
> **Response to R3**
>
> R3.1. **”I suggest the authors consider adding the CommonsenseQA task.”**: Thanks for the suggestion. We did evaluate CommonsenseQA as one of the test dataset sources. However, CommonsenseQA authors have requested not to use ConceptNet and OMCS, as it have been extensively used for creation of the dataset, and hence, the commonsense knowledge has a large overlap, and would become explicit.They also have a separate leaderboard for those methods that use explicit commonsense knowledge.
>
> R3.2. **“The authors claim in the conclusion that the proposed methods "surpass complex graph-based methods"?”**: We have cited the graph based methods from the table which we outperform,  KagNet and GBR.
>
> R3.3. **“I would suggest the authors do not claim the methods are part of the novelty of the paper as they are very trivial.”**: We have rephrased our contribution to emphasize more on the analysis.
>
> R3.4. **“The experiment section does not show any qualitative analysis or case studies.”**: More details are present in the supplementary. We have moved one section to the main paper. We did analyse the quality of the retrieved knowledge sentences. Please see the new Table 5 in the main, which was present earlier in the supplementary.
>
> R3.5. **“The retrieval module can be replaced with pre-trained DPR [4], which is generally better than BM25.”**: DPR needs ground-truth relevant passages for training. For datasets like, Natural Questions, the ground-truth paragraphs are provided which can be used as positive samples. We do not possess the actual relevant knowledge for any of the tasks currently. In future, we will work on creating a ground-truth commonsense retrieval dataset containing implicit knowledge,  to enable training of such neural retriever models.

---

### Official Review · Reviewer_QRDW · 2021-07-20
**An interesting exploration of how retrieved knowledge can be useful for commonsense reasoning QA tasks**

**Rating:** 8
**Confidence:** 4

**Review:**

This paper asks how to improve commonsense question answering by infusing or providing potentially useful knowledge to BERT-style models. Authors explore using finetuning on and/or retrieving of useful or related knowledge for questions, and experiment with multiple ways to combine the useful knowledge with the questions. Results show that simultaneously finetuning on related knowledge source and retrieving a few examples for each question improves performance the best.

The paper is well-executed, provides convincing experiments and analyses, and yields interesting results about the usefulness of external-ish knowledge.

Some questions or comments:
- According to the paper, the finetuning is done on both the MLM and the NSP objectives; however, the RoBERTa paper showed that the NSP objective isn't that useful, and wasn't used to train RoBERTa. I believe the results will not change that much, but it would probably be better if authors re-ran their experiments only using the MLM objective.
- The method section is a little hard to follow; I wish the authors had included a high-level modelling diagram that illustrates how each of the {concat, max, ...} strategies work exactly, and how they are combined to feed into BERT/RoBERTa.
- [Minor] the introduction mentions the WSC example and some retrieved passage that is supposedly helpful to answer the WSC example. I didn't fully understand why it was the same kind of reasoning, so it would be useful to point out directly why the example is useful.

---

> ### Author Response · Authors · 2021-07-30
> **Response to R2**
>
> R2.1. **“According to the paper, the finetuning is done on both the MLM and the NSP objectives”**: We follow the protocols defined for pre-training in BERT and RoBERTa respectively. We do MLM and NSP only for BERT, and only MLM for RoBERTa. We have updated it in the paper.
>
> R2.2. **”The method section is a little hard to follow”**: Thanks for your suggestion! We have added a high level diagram.
>
> R2.3. **“[Minor] the introduction mentions the WSC example and some retrieved passage”**: We have added an explanation of the necessary reasoning needed to answer the question. The reasoning from the retrieved knowledge is "the person who is refusing, is the one who is fearing."

---

### Official Review · Reviewer_CJ6w · 2021-07-22
**Interesting experiments, but containing some overly strong claims**

**Rating:** 7
**Confidence:** 4

**Review:**

This paper presents several experiments that make use of combined knowledge from language models and unstructured knowledge bases. The paper is mostly well-written, though some choices in presentation made it difficult to follow (leading me to go back and forth). I will provide some suggestions to improve this below. The paper also contains some claims that, in my opinion, are not supported by the evidence. Since the paper does not need these claims in order to provide a nice contribution, I suggest they be adapted.

The claims of which I am not convinced based on this research are the following:

- the claim that the results show the models capability of reasoning with implicit knowledge.
=> what it means for a model to do this is a very complicated question and requires much more reflection on what implicit knowledge is and what reasoning with it means (and how that can be shown). I do not believe that solving something despite lack of lexical overlap provides evidence for this, since language models are known to capture similarity in meaning.
- "The significant improvement over BERT (3-14%) shows the ability for these methods to utilize implicit knowledge, which is especially useful for low-resource languages, target domains where we can pre-train using fewer data and use ad-hoc knowledge to solve a target task and have smaller vocab and params."
=> BERT is still huge and the quality of similar models for real low-resource languages will be much lower. For such a claim, I would expect experiments that play with very small language models and limited training data (because this also tends to be the case for low-resource languages). What is more relevant from these results is that indeed the results with BERT come closer to those of RoBERTa and that this is therefore potentially interesting for reducing training costs. It also leads to the future research question of what size vs knowledge is doing (this current work does not dive into this question, but it seems the most logical thing to examine: what mistakes does each method still make and are they similar).

As mentioned earlier, I think the paper provides sufficient contributions without these claims and I therefore suggest they be revised.

Suggestions for improvement on the presentation.

About the structure:

- the choice of spreading the general approach over various sections (Section 2 about the sets, Section 3 about the knowledge sources and their preparation and Section 4 with the rest of the method made the overall setup more difficult to follow. The first sentence in Section 4 illustrates this (it is starting with `After' which shows there is a strong connection with the previous sections). I suggest merging the three sections and use the saved space to take the reader a bit more by the hand (replacing the headers by short motivations).
- I almost missed that both validation and a separate test set were present since only the size of the validation set was originally reported and had already made a note about using separate validation for tuning: this could be prevented by introducing all three components of the data right away and clearly stating how each set is used.
- I am still a bit confused about what has actually been tested to support the hypotheses and this is largely due to the use of the term `baseline' for what, if I understand correctly turn out to be the actual experiments. I recommend removing the full table for tuning and use the additional space to explain better what the reported results are.
- The paper states "As a result, the knowledge does not overlap with the gold answer, making the task trivial." This statement (in particular the choice of the word ``trivial'') can be a bit confusing, since trivial is also often used to state something is extremely obvious. I assume the problem is in fact the opposite; that using knowledge correctly would lead to the wrong outcome and that therefore the outcome on these instances do not contribute to insights on the methods?

---

> ### Author Response · Authors · 2021-07-30
> **Response to R1**
>
> R1.1. **"the claim that the results show the models capability of reasoning with implicit knowledge."**: We understand the concern. We meant given our setup of commonsense QA with implicit retrieved knowledge sentences, there is a significant improvement in the performance. As these datasets require commonsense reasoning, this improvement can be attributed to the increased ability to reason with such implicit knowledge sentences as guidance. We have updated the section accordingly.
> We have studied how the model behaves for different types of knowledge, i.e, explicit, implicit and irrelevant. This analysis is present in the supplementary material. But we don't have all knowledge variants for the same particular question. We will include an additional analysis of how the model prediction changes when we provide explicit, implicit and irrelevant knowledge to better understand if the model is able to answer correctly for both the explicit and implicit cases.
>
> R1.2. **"The significant improvement over BERT (3-14%) shows the ability for these methods to utilize implicit knowledge, which is especially useful for low-resource languages”**: We agree that compared to methods that did not use any pre-training on any source of text present online, BERT is indeed a larger model. However, BERT was trained only with 13GB data, v/s  160 GB for RoBERTa, and GPT-3, T5 and all using C4 and super-massive datasets (> 1TB). We agree that for very low resource languages that do not even possess ~10GB of data for pre-training a BERT model, it is difficult. We have added this perspective and limitation. We agree the proposed experiments are better suited for the low-resource setting.
>
> R1.3. **“the choice of spreading the general approach over various sections”**: Thanks for the suggestion. We have incorporated reviewer suggestions to the best we could during the limited time. We will address this in the final camera ready version.
>
> R1.4. **“I almost missed that both validation and a separate test set were present”**: Test set samples are blind for these 3 challenge datasets. We uploaded our model to the evaluation server to get test set scores. We have mentioned this in the relevant section.
>
> R1.5. **“I am still a bit confused about what has actually been tested to support the hypotheses”**: We used "baseline" for the 3 methods, Concat, Max, Simple-Sum as they are stronger methods compared to the simple-baseline of no knowledge. We have clarified this in the introduction and the experiments section.
>
> R1.6. **“The paper states "As a result, the knowledge does not overlap with the gold answer, making the task trivial."”**: We have rephrased this sentence. We meant, the knowledge does not overlap with the gold answer, because if it did significantly overlap, the model will use lexical overlap as a hint to solve the task instead of performing the intended reasoning.

---

### Author Response · Authors · 2021-07-30
**General Response**

We are thankful to the reviewers for their insightful feedback. We are encouraged that reviewers found our paper to be well-written (R1), well-executed and solid work (R2, R3), has sufficient contributions (R1) and interesting experiments/exploration/results (R1, R2, R3), convincing and detailed empirical analyses (R2, R3), builds an experimental framework (R3). We have incorporated reviewer suggestions to the best we could during the limited time. We will address the remaining later in the final camera ready version. We answer specific concerns to each of the reviewers.

---

### Decision · Program_Chairs · 2021-08-17

**Decision:**

Accept

**Comment:**

This paper proposes new ways to combine pretrained models with external knowledge for commonsense reasoning tasks. All the reviewers thought the paper was well-executed and interesting. One reviewer raised some concerns about overclaiming and paper organization, which the authors have promised to address.